# Hepatic Peroxisome Proliferator-Activated Receptor Alpha Dysfunction in Porcine Septic Shock

**DOI:** 10.3390/cells11244080

**Published:** 2022-12-16

**Authors:** Jolien Vandewalle, Bruno Garcia, Steven Timmermans, Tineke Vanderhaeghen, Lise Van Wyngene, Melanie Eggermont, Hester Dufoor, Céline Van Dender, Fëllanza Halimi, Siska Croubels, Antoine Herpain, Claude Libert

**Affiliations:** 1VIB Center for Inflammation Research, VIB, 9052 Ghent, Belgium; 2Department for Biomedical Molecular Biology, Faculty of Sciences, Ghent University, 9052 Ghent, Belgium; 3Experimental Laboratory of Intensive Care, Université Libre de Bruxelles, 1050 Brussels, Belgium; 4Department of Intensive Care, Centre Hospitalier Universitaire de Lille, 59000 Lille, France; 5Department of Pathobiology, Pharmacology and Zoological Medicine, Faculty of Veterinary Medicine, Ghent University, 9820 Merelbeke, Belgium; 6Department of Intensive Care, Erasme University Hospital—HUB, Université Libre de Bruxelles, 1050 Brussels, Belgium; 7Department of Intensive Care, St.-Pierre University Hospital, Université Libre de Bruxelles, 1050 Brussels, Belgium

**Keywords:** sepsis, swine, metabolism, PPARα, free fatty acids

## Abstract

Despite decades of research, sepsis remains one of the most urgent unmet medical needs. Mechanistic investigations into sepsis have mainly focused on targeting inflammatory pathways; however, recent data indicate that sepsis should also be seen as a metabolic disease. Targeting metabolic dysregulations that take place in sepsis might uncover novel therapeutic opportunities. The role of peroxisome proliferator-activated receptor alpha (PPARα) in liver dysfunction during sepsis has recently been described, and restoring PPARα signaling has proven to be successful in mouse polymicrobial sepsis. To confirm that such therapy might be translated to septic patients, we analyzed metabolic perturbations in the liver of a porcine fecal peritonitis model. Resuscitation with fluids, vasopressor, antimicrobial therapy and abdominal lavage were applied to the pigs in order to mimic human clinical care. By using RNA-seq, we detected downregulated PPARα signaling in the livers of septic pigs and that reduced PPARα levels correlated well with disease severity. As PPARα regulates the expression of many genes involved in fatty acid oxidation, the reduced expression of these target genes, concomitant with increased free fatty acids in plasma and ectopic lipid deposition in the liver, was observed. The results obtained with pigs are in agreement with earlier observations seen in mice and support the potential of targeting defective PPARα signaling in clinical research.

## 1. Introduction

Sepsis 3.0 defines sepsis as a life-threatening organ dysfunction caused by a dysregulated host response to infection [1]. Sepsis is the most important cause of morbidity and mortality in human patients admitted to intensive care units (ICU), with a significant cost impact in health care worldwide. Recent estimates suggest a yearly burden of 48.9 million sepsis cases and 11 million deaths worldwide [2]. After five decades of research, no innovative drugs addressing mechanistic pathways have become available to treat septic patients. Sepsis is classically considered as an overreaction of the immune system. However, anti-inflammatory drugs have failed in clinical trials [3]. Current management of sepsis is supportive rather than curative and essentially relies on antibiotics to eradicate the bacterial infection, fluid resuscitation with a vasopressor to maintain an adequate tissular perfusion and mechanical support for organs at risk of failing [4,5]. Recent studies have described the possible role of metabolic changes that take place in the liver during sepsis, and targeting these alterations holds much interest for the management of sepsis [4,6].

One of the promising targets is peroxisome proliferator-activated receptor alpha (PPARα) signaling. PPARα is a 52 kDa protein highly expressed in metabolic tissues and activated by free fatty acids (FFAs) and other lipid derivatives. As a member of the nuclear receptor family, PPARα has a conserved modular structure consisting of an N-terminal domain, important for transcriptional activation, a DNA-binding domain that contains zinc fingers, a short hinge region and the C-terminal ligand-binding domain [7,8]. Upon activation, PPARα regulates gene transcription by forming a heterodimer with retinoid X receptor and binding to specific DNA sequences referred to as PPARα response elements (PPRE). PPARα regulates a wide variety of genes, of which most are involved in diverse aspects of lipid metabolism, including FFA transport, oxidation of FFAs and ketogenesis (for example *ACOX1*, *HMGCS2*, *CPT1* and *CPT2*). In this way, PPARα orchestrates metabolic homeostasis during energy deprivation, the latter leading to the release of FFAs from white adipose tissue [7,8].

Sepsis is typically associated with the induction of a starvation response due to an increased energy need on the one hand, and anorexia (defined by a lack of food intake and/or appetite) on the other [6,9,10]. Despite the essential role of PPARα to cope with starvation, recent studies in murine sepsis have found that PPARα expression dramatically declines and hence loses function in the liver [11,12]. This results in impaired lipid metabolism, coinciding with ectopic lipid accumulation, lipotoxicity and increased mortality. Preventing PPARα dysfunction reduces sepsis mortality, making this pathway an interesting target to evaluate in clinical research [11,12]. The above-mentioned studies were performed on mice, but it remains to be confirmed whether these observations also occur in human sepsis. 

Rodents are used in 94% of all sepsis studies because of their low cost, well-characterized genome and the opportunity to generate and use transgenic strains [13]. However, mice are less suitable as a model for sepsis due to their high resilience to infection, different pathophysiology to humans and technical constraints for performing a goal-orientated resuscitation and source control [14]. Therefore, validation of mice preclinical findings in a larger mammalian animal model enhances the translational potential towards human sepsis drastically [13]. Pigs are appropriate animal models to close the gap between rodent and human studies given their homology in size, physiology and pathophysiology, especially when studying metabolic and infectious diseases [15,16]. 

In this current study, we report that sepsis-induced changes in the liver, as observed in mice, are actually also relevant in a porcine peritonitis model. Based on bulk RNA sequencing (RNA-seq), we found PPARα dysfunction and concomitant lipid accumulation in the blood and liver of septic pigs in direct relation to the degree of hemodynamic alterations in the animals. Our results strengthen the potential validity of poor PPARα signaling as a new therapeutic target for human sepsis patients. 

## 2. Materials and Methods

### 2.1. Experimental Procedure

Fecal peritonitis resulting in septic shock was introduced in nine pigs (*Sus scrofa domesticus*, RA-SE Genetics, Belgium, weighing ± 50 kg) (5 ♂ and 4 ♀) by intraperitoneal instillation of 3 g/kg of autologous feces previously collected from the animal’s enclosure and diluted in 300 mL of 5% glucose solution. As a control, three sham pigs (2 ♂ and 1 ♀), consisting of anesthesia and surgical preparation but without sepsis induction, were applied. Animals were fasted for 18 h prior to the start of the experiment, with free access to water. All the septic animals developed shock (mean arterial pressure (MAP) ≤ 50 mmHg)—within 5.9 ± 1.4 h after the onset of sepsis—and were then left untreated for 1 h to consolidate multi-organ dysfunction. Resuscitation fluids, norepinephrine (NE), antibiotics treatment and abdominal lavage were applied during the following 8 h to mimic human clinical cares, after which the animals were euthanized for organ isolation (liver and plasma). The study protocol for the pigs followed the EU Directive (2010/63/EU) for animal experiments and was approved by the local animal ethics committee (Comité Ethique du Bien-Être Animal; protocol number 724N) from the Université Libre de Bruxelles (ULB) in Brussels (Belgium). Pig experiments were performed in the Experimental Laboratory of Intensive Care of the ULB (LA1230406) and under the ARRIVE guidelines and MQTiPSS recommendations. Analysis of the samples was performed in the Inflammation Research Center (IRC) in Zwijnaarde. A detailed description of the experimental procedure to induce septic shock in the pigs is provided in [17]. Samples were isolated from the NE group at the end of the experiment (vasopressor 2 timepoint). Note, we isolated one extra pig that was used for a pilot experiment.

### 2.2. Biological Samples

Systemic blood samples for FFA analysis were collected from the femoral artery at the end of the experiment and immediately centrifuged, and plasma was frozen at −20 °C until further processing. FFAs were determined via colorimetric assays according to the manufacturer’s instructions (KA1667, Abnova, Taipei City, Taiwan).

### 2.3. Liver Transcriptomic Analysis

Liver biopsies taken from the fourth segment were isolated and stored in RNALater (AM7021, Invitrogen, Waltham, MA, USA). RNA was isolated using the Aurum total RNA mini kit (732-6820, Biorad, Temse, Belgium). RNA quality was checked with the Agilent RNA 6000 Pico Kit (Agilent Technologies, Santa Clara, CA, USA). The RNA was used for creating an Illumina sequencing library using the Illumina TruSeqLT stranded RNA-seq library protocol (VIB Nucleomics Core, Belgium) and paired-end sequencing (2 × 150 bp) was done on an Illumina Novaseq 6000. The obtained reads were mapped to the pig (*Sus scrofa*, Sscrofa11.1) reference transcriptome/genome with hisat v2.0.4 [18]. Gene-level read counts were obtained with the feature Counts software (part of the subread package) [19]. Multimapping reads were excluded from the assignment. Differential gene expression was assessed with the DESeq2 package [20], with the FDR set at 5%. Motif finding for multiple motifs, or de novo motif finding, was performed using the HOMER software [21]. We used the promoter region (start offset: −500 bp, end offset: 50 bp downstream of transcription start site TSS) to search for known motif enrichment and de novo motifs. Gene ontology (GO) term enrichment on selected groups of genes was performed via the Enrichr tool [22].

### 2.4. LipidTOX Staining

Liver pieces (~1 cm^3^) were isolated and stored in cold antigenfix for 1–2 h. Then, the liver was washed twice in cold PBS and put in 34% sucrose at 4 °C while constantly agitating for at least 10 h and up to 24 h. The next day, liver pieces were stored in NEC50 and frozen at −80 °C until further processing. Cryostat sections of 20 μm thickness were rehydrated in PBS for 5 min after which the sections were blocked in blocking buffer (2% BSA, 1% fetal calf serum, 1% goat serum, in 0.5% saponin) for 30 min at RT. The antibody mix (LipidTOX Deep Red (1:400, Life Technologies Europe B.V., Merelbeke, Belgium); Acti-stain 488 Phalloidin (1:150, PHDG1, Cytoskeleton Inc., St. Denver, CO, USA)) was added and incubated for 2 h at RT. After washing with PBS for 5 min, nuclear staining (Hoechst (1:1.000, Sigma-Aldrich N.V., Hoeilaart, Belgium)) was added for 5 min at RT. Slides were washed in PBS for 5 min, quickly rinsed in water to remove residual salt and mounted. For each cryosection, eight Z-stacks of 3 areas per coupe were imaged with a spinning disk confocal microscope (Zeiss, White Plains, NY, USA), using a 40× Plan-Apochromat objective lens (1.4 Oil DIC (UV) VIS-IR M27, Zeiss)) at a pixel size of 0.167 μm and at optimal Z-resolution (240 mm). Z-stacks were processed in Volocity (PerkinElmer, Waltham, MA, USA) and the amount of lipid droplets was calculated. We had two biological samples per group (sham vs. sepsis). Two liver pieces from each pig were isolated and 3 areas per liver piece were analyzed for technical variance. The average was taken of the six technical repeats per animal and the biological data was used for analyses.

### 2.5. Datasets and Databases

RNA-seq data of the pigs are deposited at the National Center for Biotechnology Information (NCBI) Gene Expression Omnibus public database (http://www.ncbi.nlm.nih.gov/geo/) under accession number GSE218636 (accessed on 23 November 2022). PPARα responsive genes are retrieved from the publicly available dataset deposited at the NCBI under accession number GSE139484 (accessed on 23 November 2019). PPARα responsive genes are considered as those responsive to the PPARα agonist GW7647 in sham condition. Genes involved in FA oxidation are retrieved from MGI (http://www.informatics.jax.org/vocab/gene_ontology/) with GO:0019395 (accessed on 18 October 2022). 121 genes are involved in this pathway of which 87 are found in pigs.

### 2.6. Statistics

Figures are generated in Graphpad prism 9. When comparing sham versus sepsis data, a Student’s *t*-test was used in an unpaired two-tailed fashion. Mean ± SEM is shown in bar graphs. Violin plots were analyzed with a Wilcoxon signed-rank test. 

## 3. Results

### 3.1. Hepatic PPARα Dysfunction in Porcine Septic Shock

#### 3.1.1. Inflammation and Metabolic Dysregulation in Liver upon Septic Shock

All the septic animals developed shock (mean arterial pressure (MAP) ≤ 50 mmHg)—within 5.9 ± 1.4 h after the onset of sepsis—with severe tissular hypoperfusion (Figure 1). Septic shock induced an hyperdynamic cardiovascular response, restored by the introduction of fluid expansion and subsequent NE infusion, which normalized tissular perfusion for the entire period of septic shock resuscitation. Multiple organ failure was observed, with respiratory and renal dysfunction, on top of circulatory failure and severe capillary leakage. Clinical parameters measured at the timepoint just before euthanizing are summarized in Table 1. A more comprehensive description of the model severity and the response to resuscitation care can be found in the princeps study publication [17].

Relative differences in transcriptional changes observed in the liver upon septic shock were visualized with principal component analysis (PCA) and a clear separation of sham versus sepsis samples was observed (Figure 2A). Sepsis induced significant upregulation of 1892 genes and downregulation of 2043 genes in the liver of the pigs (*p* < 0.05) compared to sham animals (Figure 2B). A list of differentially expressed genes and their expression level is included in the supplementals (Appendix A). To obtain information on the processes that were specifically induced or inhibited in the septic pigs by these upregulated and downregulated genes, they were analyzed via gene set enrichment against various libraries using Enrichr. Genes upregulated in the liver upon sepsis were associated with the immune system, innate immune system and neutrophil degranulation (Figure 2C). These are typical pathways activated upon inflammation. The downregulated genes were associated with metabolism, more specifically metabolism of lipids and FFAs, as well as amino acids and steroids (Figure 2D). As targeting the inflammatory pathway in sepsis trials did not result in successful therapeutic targets [3] and based on our previous studies in murine sepsis [12], we focused on the pathways that are predicted to be downregulated upon sepsis. Restoring downregulated pathways that normally have a protective function could provide novel therapeutic opportunities.

#### 3.1.2. PPARα Dysfunction

Analysis of the downregulated genes with Wiki pathway analysis identified PPARα signaling as most affected pathway during sepsis (Figure 3A). Motif analysis by HOMER of the downregulated genes (*p* < 0.05 and with a mouse orthologue) upon sepsis revealed a PPARα motif in the top 5 of the known motifs (Figure 3B). The mean log fold change (LFC) upon sepsis of the 293 downregulated genes that contain a PPRE identified by HOMER was −1.8 (Appendix A). This PPARα motif was not retrieved in the motifs of the upregulated genes (*p* < 0.05 and with a mouse orthologue), the top motifs of which were generally associated with the inflammatory response (Figure 3B), e.g., the NF-κB DNA binding motif. To study the effect of PPARα loss of function genome wide, we plotted the LFC upon sepsis of all PPARα-induced genes (Appendix B). The median LFC of these genes was −0.27 and significantly differed from the baseline LFC 0 (Figure 3C), which confirms that PPARα appears significantly less active as a transcription factor in sepsis than sham pigs. As PPARα is involved in FFA oxidation, the LFC of all genes involved in FFA oxidation based on gene ontology analysis (GO:0019395), and which depend on PPARα for their expression, were compared between sham and sepsis samples. The median LFC of these genes was −1.07 (Figure 3D) (Appendix A). As an illustration, we plotted the expression level of PPARα and several PPARα targets involved in β-oxidation and ketogenesis to compare the expression in septic livers to those in sham condition (Figure 3E). The mRNA expression of PPARα in septic animals was only 36% of that in sham animals. Likewise, genes involved in the FFA β-oxidation pathway were reduced by more than half upon sepsis. Interestingly, PPARα levels in the liver correlated negatively with the NE requirement of the septic pigs (Figure 3F). This correlation implies that pigs with lower PPARα levels have a higher need for NE infusion to maintain their blood pressure, which is indicative of a more severe sepsis. Together, these data support the notion that PPARα signaling is dysfunctional in the liver of septic pigs and correlates with sepsis severity.

#### 3.1.3. Increased FFA in Blood and Ectopic Lipid Accumulation in Liver

Since PPARα is the major transcription factor involved in β-oxidation of FFAs, and since we observe significant problems with the expression of PPARα and genes involved in FFA β-oxidation, we hypothesized that FFAs will accumulate in the blood due to processing problems. Indeed, total FFAs were increased significantly in the pig’s blood upon sepsis (Figure 4A). When FFAs are high in circulation, the liver sequestrates them in lipid droplets [12]. To investigate whether ectopic lipid accumulation is occurring in the liver of septic pigs, cryosections of liver were stained with LipidTOX, a fluorescent dye with high affinity for neutral lipids. Extensive lipid droplet accumulation was observed in the livers of septic animals, whereas this was only minimal in sham samples (Figure 4B,C), indicating more lipid accumulation in the liver of septic subjects.

## 4. Discussion

In contrast to the Sepsis 2.0 definition, the latest definition of sepsis no longer refers to inflammation as an essential pathway [1]. One of the reasons for this adjustment is that clinical trials targeting the inflammatory pathway have failed to provide significant survival benefits [3]. Potential clarifications for this failure might be the use of inappropriate animals as preclinical models. Given the complex nature of sepsis, it is unlikely that findings based on one species will be able to mimic all aspects of the clinical and biological complexity observed in human sepsis. Therefore, it is advisable to confirm key findings in other mammals, such as pigs [23].

Next, it has become increasingly clear that other pathways, such as certain metabolic pathways, are playing a crucial role in the pathogenesis of sepsis [4]. A large proteomic and metabolic screen on the plasma of human sepsis patients identified glucose metabolism and FFA β-oxidation pathways as being significantly different between sepsis survivors and non-survivors [24]. Moreover, these pathways differed consistently among several sets of patients, and diverged more as death approached [24]. FFA β-oxidation is a multistep process that breaks down FFAs in the mitochondria to produce energy. In times of starvation, lipolysis liberates FFAs from adipose tissue to foresee in energy [25]. In septic subjects, lipolysis is observed as a way to cope with the negative energy balance that is the result of an increased energy need and a decreased food intake during sepsis [9,10,26]. However, we recently observed that the transformation of FFAs to acetyl-CoA and ketones fails due to defects in PPARα signaling, leading to an accumulation of these substrates. Indeed, already 6 h after the onset, FFAs increase in the blood of septic mice [12]. Similarly, FFAs increase in the blood of septic patients and these levels correlate well with the clinical severity [12]. Mice with a deletion of PPARα in their hepatocytes display an increased mortality upon sepsis [27]. Preventing PPARα dysfunction with pemafibrate, however, reduces sepsis mortality. Pemafibrate is a novel selective PPARα modulator (SPPARMα) that restores PPARα expression, thereby decreasing FFA plasma levels and lipid accumulation in the liver [12]. Together these data reveal PPARα as an interesting new target for sepsis.

The role of PPARα in human sepsis patients has already been investigated. In the blood of septic children, the decreased expression of PPARα responsive genes was observed, with lower levels being associated with a more severe disease status [28]. Nevertheless, the major FFA-metabolizing organ which depends on PPARα is the liver. Determination of hepatic PPARα levels and function in human sepsis patients is, however, challenging, primarily due to the coagulation problems in sepsis patients. Post-mortem liver biopsies could identify lower PPARα levels in non-surviving critically ill patients [27]; however, these results should be interpreted with caution as RNA quality rapidly declines after death [29].

We studied the liver of septic pigs in order to minimize the gap between mouse and human research. In the current study, we analyzed the liver of septic pigs, since the liver plays a key role in metabolic rearrangements. In accordance with the literature [30], upregulated genes in porcine livers upon sepsis are generally associated with inflammatory pathways, whereas downregulated genes are linked to metabolic pathways. Interestingly, the pathway affected with the highest probability is PPARα signaling. PPARα targets related to FFA β-oxidation are especially reduced in the livers of sepsis pigs, which might explain the increased FFA levels in plasma and lipid accumulation in their livers. In murine sepsis, reduced mRNA expression of PPARα and its targets involved in FFA β-oxidation could indeed be linked to a reduced capability of liver explants to metabolize palmitic acid ex vivo, as measured with the Seahorse technology [12].

The strength of this current study is the use of an advanced ICU animal model that follows the MQTiPSS recommendations (including abdominal lavage, antimicrobial therapy and early vasopressor introduction) with the aim of improving the clinical translatability of experimental findings. A downside of doing this type of research, especially when using pigs, is the high cost in comparison to research using rodents as the animal model. This downside can be countered by performing interdisciplinary or multicenter studies involving the isolation of several tissues and organs simultaneously. The collaborative study reported here is an example. Liver biopsies and plasma were isolated from septic shock pigs that were also used in a study on the myocardial effects of vasopressors [17]. Due to logistical reasons, we were, however, not able to quantify FFA β-oxidation ex vivo in the pig samples, so conclusive proof of the link between PPARα resistance on the one hand and FFA increase and lipotoxicity on the other is lacking in pigs. Moreover, a determination of mitochondrial number might be interesting, as the number of mitochondria can also influence FFA β-oxidation capacity in cells. Lastly, the effect of the PPARα agonist pemafibrate on hepatic PPARα function, lipotoxicity and survival still needs to be determined to solidify the proposed hypotheses.

Collectively, these data suggest that problems with PPARα expression and activity in the liver might contribute to problems with β-oxidation of FFAs. This in turn leads to deposition of lipid storages in the liver upon sepsis. The data obtained in porcine septic subjects thus confirm the results obtained in murine sepsis and support the potential of targeting defective PPARα signaling in the clinic. Elucidation of the upstream signal(s) causing PPARα dysfunction might uncover novel therapeutic opportunities in sepsis. Moreover, clinical trials are warranted to validate the therapeutic applicability of this axis (i.e., PPARα resistance–FFA increase–lipotoxicity) in human individuals with sepsis.

## Figures and Tables

**Figure 1 cells-11-04080-f001:**
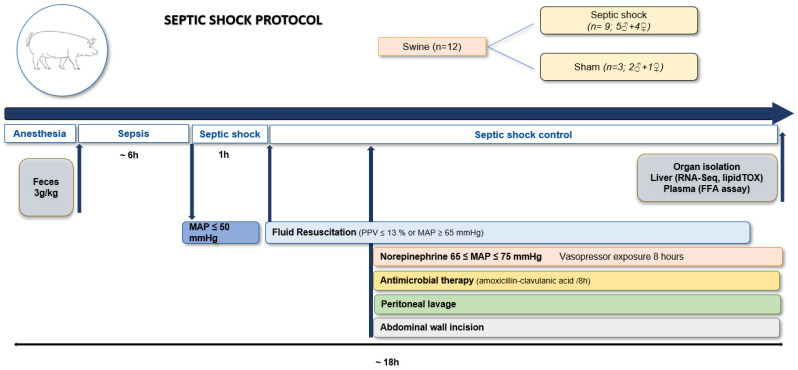
Protocol timeline. Animals were allowed to develop sepsis until a severe hypotension with mean arterial pressure (MAP) ≤ 50 mmHg was obtained. Severe hypotension (MAP between 45 and 50 mmHg) was allowed for one hour. Thereafter, full resuscitation was started, aiming to restore the pulse pressure variation (PPV) to ≤13% or MAP ≥ 65 mmHg. After achieving this objective, antibiotics and abdominal lavage were applied to control the infection. Norepinephrine was added to the infusion to maintain a MAP of 65–75 mmHg for 8 h. After this timepoint, the animals were euthanized to collect liver and plasma for subsequent analysis (i.e., ~18 h after the start of the experiment).

**Figure 2 cells-11-04080-f002:**
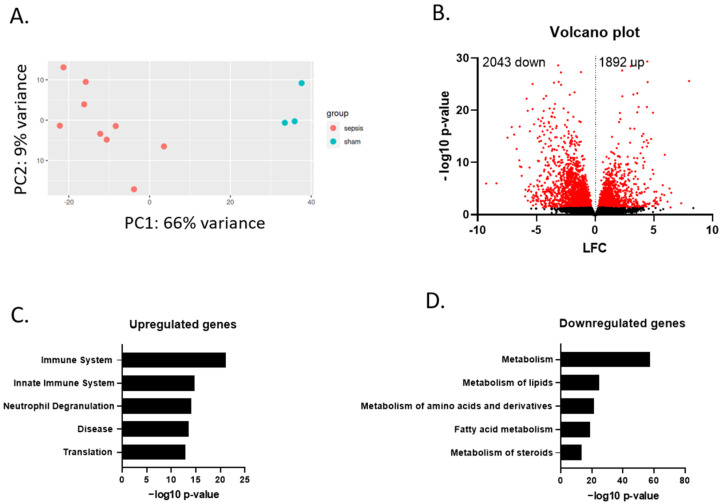
Metabolic dysregulation in livers of porcine sepsis. (**A**) PCA plot of overall gene expression in liver samples from sham and sepsis pigs. PCA plot clarifies the variance within samples per condition. (**B**) Volcano plot with differential expression of genes affected by sepsis compared to sham, as measured by bulk RNA sequencing. Genes significantly affected are indicated with red dots (*p* < 0.05). Genes with −log10 *p*-value above 30 are omitted for clarity of the plot. (**C**) Top five enriched gene ontology (GO) terms for genes that are upregulated upon sepsis (*p* < 0.05). (**D**) Top five enriched GO terms for genes that are downregulated upon sepsis (*p* < 0.05). Analyses of (**C**,**D**) was performed with the Enrichr tool.

**Figure 3 cells-11-04080-f003:**
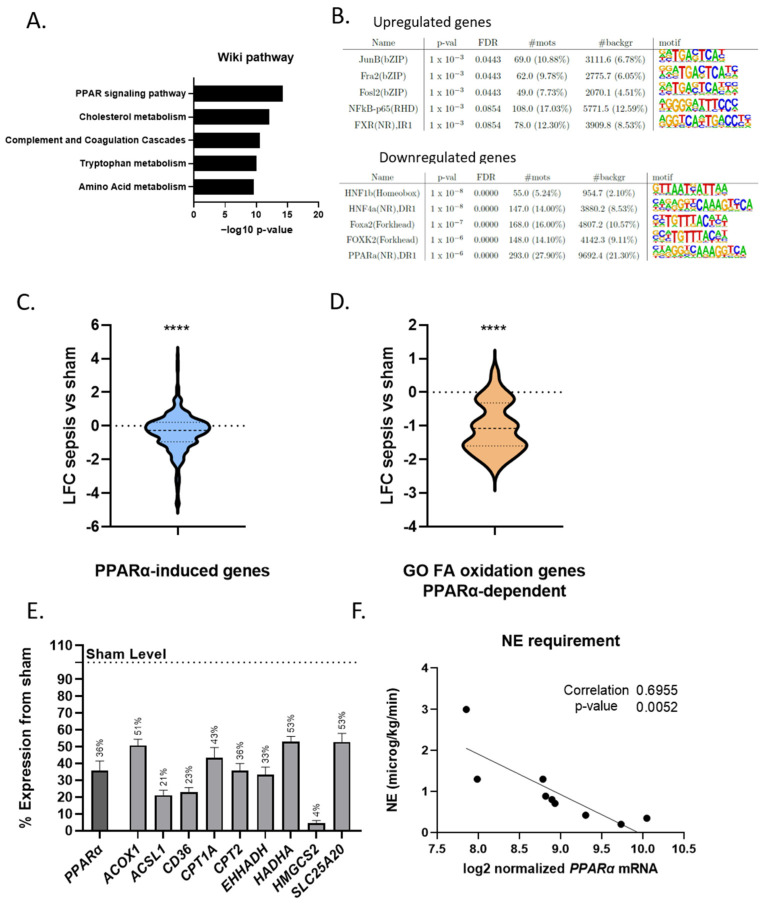
PPARα dysfunction in livers of porcine sepsis. (**A**) Top five enriched gene ontology (GO) terms for genes that are downregulated upon sepsis (*p* < 0.05). Analysis was performed with the Wiki pathway tool. (**B**) Homer motif analysis to detect DNA binding motifs in the 500 base pairs upstream of the transcription start site of up- and down regulated genes upon sepsis (*p* < 0.05 and for genes with a mouse orthologue). Top five motifs ranked according to *p*-value is shown. (**C**) Violin plot showing LFC upon sepsis of genes that are known to be induced by the PPARα agonist GW7647 in murine sepsis (*p* < 0.05). LFC of the porcine genes with a mouse orthologue upon sepsis is displayed (389 genes). Median and upper and lower quartiles are shown with a horizontal dotted line in the violin plot. Median is significantly different from baseline LFC 0. **** *p* < 0.0001. (**D**) Violin plot showing LFC upon sepsis of genes involved in FA oxidation (GO00193951, 121 genes) that are known to be induced by the PPARα agonist GW7647 (34/121) in murine sepsis (*p* < 0.05). LFC of the porcine genes with a mouse orthologue upon sepsis is displayed (29 genes). Median and upper and lower quartiles are shown with a horizontal dotted line in the violin plot. Median is significantly different from baseline LFC 0. **** *p* < 0.0001. (**E**) Expression level of PPARα and genes involved in β-oxidation of FFAs and ketogenesis is displayed. The expression level of sham pigs is set as 100% and compared to the level in sepsis pigs. (**F**) Correlation curve plotting the norepinephrine (NE) requirement of each septic pig to their PPARα mRNA level in the liver (RNA-seq counts are log2 normalized). Data is analyzed with a simple linear regression. R^2^ depicts correlation of 0.6955. The slope is significantly different from zero. N = 9.

**Figure 4 cells-11-04080-f004:**
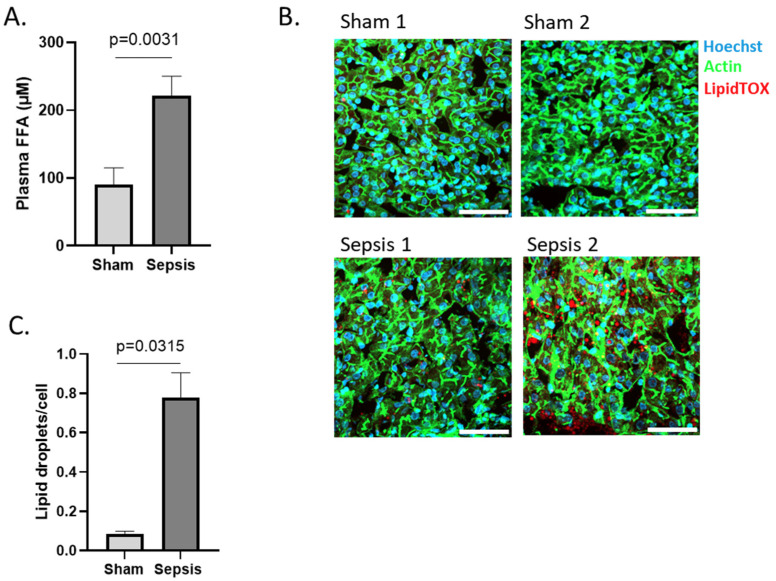
Metabolic disease parameters in blood and liver of porcine sepsis. (**A**) Concentration FFA in plasma of sham (n = 3) and sepsis (n = 9) pigs. (**B**) Immunofluorescent images of porcine liver after sham or sepsis. Cryosections were stained with Hoechst (blue), acti-stain (green) and lipidTOX (red). Z-stacks (8/region) were generated in 3 areas scattered across the entire tissue section. White scale bar = 50 μm. One representative picture of each biological repeat is shown. (**C**) The amount of lipid droplets (LDs)/cell were calculated for each Z-stack. Averages of the amount of the LDs were converged for each subject and biological replicates (n = 2) are used in the figure.

**Table 1 cells-11-04080-t001:** Clinical parameters determined in pigs just before euthanizing.

		Sham	Septic Shock	*p*-Value
**Hemodynamic parameters**	MAP (mmHg)	68 ± 2	68 ± 3	0.85
	HR (/min)	87 ± 3	155 ± 21	0.03
	CO (L/min)	6 ± 1	9 ± 2	0.03
	NE (μg/kg/min)	0 ± 0	1 ± 0.84	0.08
	SvO_2_ (%)	64 ± 3	73 ± 5	0.02
	PCO_2_ gap (mmHg)	5 ± 1	5 ± 5	0.98
**Respiratory function**	RR (/min)	17 ± 3	20 ± 3	0.10
	Tidal Volume (mL)	375 ± 35	385 ± 50	0.76
	PaCO_2_ (mm Hg)	52 ± 1	47 ± 5	0.15
	PaO_2_/FiO_2_ ratio	330 ± 58	253 ± 58	0.07
**Metabolism**	T°	38.8 ± 0.8	38.8 ± 0.4	0.80
	Glucose (mg/dL)	107 ± 10	115 ± 23	0.61
	Lactate (mmol/L)	0.9 ± 0.00	2 ± 1.1	0.12
	pH	7.46 ± 0.01	7.42 ± 0.06	0.26
	Base excess (mmol/L)	12 ± 1	5 ± 5	0.03

MAP: mean arterial pressure, HR: heart rate, CO: cardiac output, NE: norepinephrine requirement, SvO_2_: mixed venous oxygen saturation, pCO_2_ gap: veno-arterial difference in CO_2_ partial pressure, RR: respiratory rate, PaCO_2_: partial pressure of carbon dioxide, PaO_2_/FiO_2_: ratio of arterial oxygen partial pressure to fractional inspired oxygen.

## Data Availability

RNA-seq data of the pigs are deposited at the National Center for Biotechnology Information (NCBI) Gene Expression Omnibus public database (http://www.ncbi.nlm.nih.gov/geo/) under accession number GSE218636 (accessed on 23 November 2022). PPARα responsive genes are retrieved from the publicly available dataset deposited at the NCBI under accession number GSE139484 (accessed on 23 November 2019). PPARα responsive genes are considered as those responsive to the PPARα agonist GW7647 in sham condition. Genes involved in FA oxidation are retrieved from MGI (http://www.informatics.jax.org/vocab/gene_ontology/) with GO:0019395 (accessed on 18 October 2022). 121 genes are involved in this pathway of which 87 genes were found in pig.

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
