# Peer review of "Hepatic Peroxisome Proliferator-Activated Receptor Alpha Dysfunction in Porcine Septic Shock"

_cells, 2022, doi:10.3390/cells11244080_

Round 1
Reviewer 1 Report
Hepatic Peroxisome Proliferator-Activated Receptor alpha dysfunction in porcine septic shock.
Jolien Vandewalle, et al.
Summary
These studies investigated how PPARa signaling is impacted in a porcine model of sepsis. Sepsis was induced in pigs and the liver was harvested at 6 hrs for RNA analysis. The investigators observed a decrease in mRNA of many genes involved in hepatic fatty acid oxidation. Furthermore, there was increased hepatic lipid accumulation and expression of FFA.
Comments
This was a well designed study with a good model of sepsis. The phenotype of sham and septic pigs was carefully assessed in Table 1. It is a strength of the investigation that it was conducted in pigs as this model has more human relevance. The results were similar to what had been observed previously in rodents, but this did not detract from the impact of the work. RNA analysis was well done with appropriate clustering of up and down regulated genes. Analysis of the lipid droplets in the liver was an asset.
If possible, the assessment of some protein changes in the liver would strengthen the manuscript. The sepsis only last 6 hrs so it raises the question of whether PPARa receptor levels have changed. Westerns of some key proteins, especially those down-regulated, would be beneficial. Also, determination of mitochondria number would be interesting although fatty acid oxidation could be elevated without change in mitochondrial number.
Minor comments
Fig 3E shows a decrease in CPT1a expression as would be expected for an enzyme that is rate limiting in FA oxidation. It was surprising that CPT1a was not included in supplemental data Table 2. I did not check for all genes. Also, CPT1 did not appear in PPARa responsive genes list in supplemental Table 3 although it is well known to be regulated by PPARa in rodents and humans. Is there an explanation for this?
Reviewer 2 Report
The paper is well-written and describes work which has been executed to a high standard. I have no hesitation in recommending publication. The methods and results are described in detail and the ethical aspects are covered. The introduction and discussion are comprehensive and the figures seem to be of high quality. .
The word 'princeps' which apparently is a latin word meaning 'principal' or 'lead' is used once, so I would recommend either using an english equivalent or italicising it.
In Fig. 1 and the legend, I was confused by the the use of +/- e.g the last phrase in brackets "(i.e.+/- 18 hours after the start of the experiment)". I am also not sure what it means in the figure itself. Do the authors mean approximately? If so a different symbol should be used.
Author Response
Please
Reviewer 2:
The paper is well-written and describes work which has been executed to a high standard. I have no hesitation in recommending publication. The methods and results are described in detail and the ethical aspects are covered. The introduction and discussion are comprehensive and the figures seem to be of high quality. .
The word 'princeps' which apparently is a latin word meaning 'principal' or 'lead' is used once, so I would recommend either using an english equivalent or italicising it.
In Fig. 1 and the legend, I was confused by the the use of +/- e.g the last phrase in brackets "(i.e.+/- 18 hours after the start of the experiment)". I am also not sure what it means in the figure itself. Do the authors mean approximately? If so a different symbol should be used.
Comments Reviewer 2:
We thank the reviewer for his/her positive comments.
- We have italicised “princeps” as the English words are not really the right words in this context (we mean the initial study where these swine subjects were used and published).
- Indeed, we mean approximately. Approximately 6hrs after the sepsis insult, the pigs start to develop septic shock, which is arbitrarily set as a mean arterial pressure (MAP) of 50 mmHg or below. This timepoint is varying between the sepsis pigs. We have used a tilde (~) as symbol for approximately in both the figure and legend.
see attachment